# Antibacterial Properties of Melanoidins Produced from Various Combinations of Maillard Reaction against Pathogenic Bacteria

Satoshi Kukuminato,[a] Kento Koyama,[a] Shigenobu Koseki[a]

[a]Graduate School of Agricultural Science, Hokkaido University, Sapporo, Japan

**ABSTRACT** Novel melanoidins are produced by the Maillard reaction. Here, melanoidins with high antibacterial activity were tested by examining various combinations of reducing sugars and amino acids as reaction substrates. Twenty-two types of melanoidins were examined by combining two reducing sugars (glucose and xylose) and eleven L-isomers of amino acids (alanine, arginine, glutamine, leucine, methionine, phenylalanine, proline, serine, threonine, tryptophan, and valine) to confirm the effects of these melanoidins on the growth of *Listeria monocytogenes* at 25℃. The melanoidins produced from the combination of D-xylose with either L-phenylalanine (Xyl-Phe) or L-proline (Xyl-Pro), for which absorbance at 420 nm was 3.5 ± 0.2, completely inhibited the growth of *L. monocytogenes* at 25℃ for 48 h. Both of the melanoidins exhibited growth inhibition of *L. monocytogenes* which was equivalent to the effect of nisin (350 IU/mL). The antimicrobial spectrum of both melanoidins was also investigated for 10 different species of bacteria, including both Gram-positive and Gram-negative species. While Xyl-Phe-based melanoidin successfully inhibited the growth of *Bacillus cereus* and *Brevibacillus brevis*, Xyl-Pro-based melanoidin inhibited the growth of *Salmonella enterica* Typhimurium. However, no clear trend in the antimicrobial spectrum of the melanoidins against different bacterial species was observed. The findings in the present study suggest that melanoidins generated from xylose with phenylalanine and/or proline could be used as potential novel alternative food preservatives derived from food ingredients to control pathogenic bacteria.

**IMPORTANCE** Although the antimicrobial effect of melanoidins has been reported in some foods, there have been few comprehensive investigations on the antimicrobial activity of combinations of reaction substrates of the Maillard reaction. The present study comprehensively investigated the potential of various combinations of reducing sugars and amino acids. Because the melanoidins examined in this study were produced simply by heating in an autoclave at 121℃ for 60 min, the targeted melanoidins can be easily produced. The melanoidins produced from combinations of xylose with either phenylalanine or proline exhibited a wide spectrum of antibiotic effects against various pathogens, including *Listeria monocytogenes*, *Bacillus cereus*, and *Salmonella enterica* Typhimurium. Since the antibacterial effect of the melanoidins on *L. monocytogenes* was equivalent to that of a nisin solution (350 IU/mL), we might expect a practical application of melanoidins as novel food preservatives.

**KEYWORDS** *Listeria monocytogenes*, phenylalanine, proline, xylose, nisin, antimicrobial activity

Various antimicrobials have been reported and used as food preservatives, extending the shelf life of processed foods by controlling bacterial growth in the final products. For example, potassium sorbate and sodium benzoate inhibit various microorganisms, such as *Vibrio parahaemolyticus*, *Bacillus mucoides*, *Bacillus subtilis*, *Staphylococcus aureus*, *Pseudomonas aeruginosa*, *Escherichia coli*, *Aspergillus flavus*, *Candida albicans*, *Fusarium oxysporum*, *Trichoderma harsianum*, and *Penicillium italicum*, although the MICs vary

Address correspondence to Shigenobu Koseki, koseki@bpe.agr.hokudai.ac.jp.

The authors declare no conflict of interest.

(1, 2). Protamine is also known as a preservative that inhibits bacterial growth (3). Furthermore, nisin, a low-molecular-weight peptide produced by *Lactococcus lactis* subsp., is widely used as a bacteriocin (4). The antimicrobial activity of nisin against Gram-positive bacteria is widely used to control *Listeria monocytogenes* because of its high sensitivity (5–7).

Although numerous antimicrobials have been reported, there are no universal antimicrobials against several microorganisms. The target bacteria of each antibacterial substance are different, and some substances do not exhibit antibacterial properties at low doses. Thus, there is still demand for novel and effective antimicrobial substances. Furthermore, antimicrobial substances which are easy to produce and have been safely eaten by humans for many years are most desirable. Under these circumstances, we focused on the antimicrobial activity of melanoidin, which is one of the final products of the Maillard reaction.

The Maillard reaction is a chemical reaction that occurs mainly during the thermal processing of foods. This multiple-step heat-promoting reaction is induced by carbonyl compounds such as reducing sugars, and amines such as amino acids, and the final product is a brown polymer called melanoidin (8). Melanoidin contributes to the quality of food because it adds flavor (9, 10), and the melanoidin produced in heated vegetables or fruits prevents the enzymatic browning reaction of endogenous phenolic compounds (11). Furthermore, melanoidin exhibits antioxidative (12–14), antihypertensive (15), prebiotic (16), and antimicrobial activities. However, since the precise structure and characteristics of melanoidin are still unclear, studies have been conducted on the identification of its chemical structure and the discovery of new substances (17, 18).

Studies on the antibacterial properties of melanoidins were first reported in the 1950s. Hachisuka et al. showed that heated glucose and amino acids inhibited the germination and/or bacterial growth of *B. subtilis*, and that the efficacy of this inhibition differed depending on the amino acid used (19). Habinshuti et al. investigated the antimicrobial activity of melanoidin produced in sunflower, soybean, and corn and determined that only sunflower-based melanoidin showed antimicrobial activity against *S. aureus* and *E. coli* O157:H7 (20).

The antibacterial activity of melanoidins derived from coffee (light and medium roast), beer (lager, abbey style, stout), and wine was investigated; in particular, coffee melanoidin (medium roast) successfully inhibited the growth of *S. aureus* (21). Peptic hydrolysates of half-fin anchovy-glucose Maillard reaction products showed a broad antibacterial spectrum (22); these products inhibited *E. coli*, *P. fluorescens*, *P. aeruginosa*, *Proteus vulgaris*, *S. aureus*, *B. subtilis*, *B. megaterium*, and *Sarcina lutea*. Moreover, the antimicrobial activity of straw-wine melanoidin was investigated, which inhibited the growth of *L. monocytogenes*, *S. enteritidis*, and *E. coli* for 18 h (23).

Although many studies on the antimicrobial activities of melanoidins have been conducted (24–32), there is still the possibility of discovering additional antimicrobial melanoidins. Because the Maillard reaction is a polymerization reaction of reducing sugars and amino acids, and there are many types of both reducing sugars and amino acids, the number of possible combinations for the reaction is enormous. Thus, unknown combinations of reducing sugars and amino acids may produce novel melanoidins with strong antibacterial activity. However, previous studies have mainly investigated the antibacterial activity of food-derived melanoidins (20–23). There has been no comprehensive investigation on the effects of combinations of reducing sugars and amino acids on the antibacterial activity of melanoidin.

The objective of the present study was to discover novel melanoidins with antibacterial activity by comprehensively examining combinations of reducing sugars and amino acids. The effectiveness of these synthesized melanoidins as antimicrobials was confirmed against a spectrum of Gram-positive and Gram-negative bacteria and compared with that of preservatives currently used in the food industry.

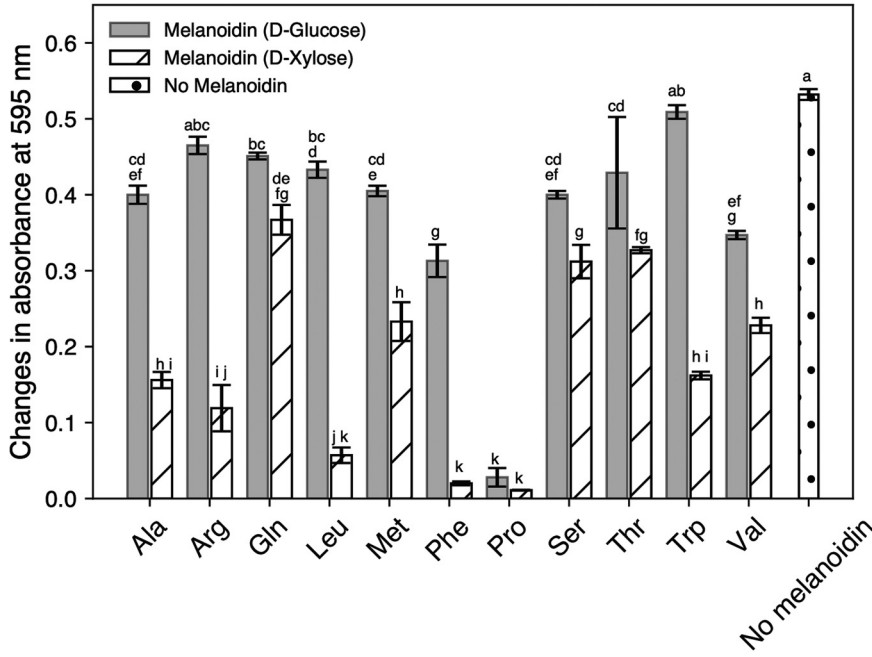

**FIG 1** Effects of the reaction substrates of melanoidin on antimicrobial activity against *L. monocytogenes*. Bacterial growth was determined by absorbance at 595 nm. Independent experiments were performed in triplicate, and the average value is shown in the bar plot. Error bars corresponds to standard deviations as determined three times. Absorbance changes over 24 h for each of the melanoidins and the negative control were compared using the Tukey-Kramer multiple range test. *P* values of the results was used for evaluation of significant difference (*P* < 0.05); values with different letters represent significant differences.

## RESULTS

**Screening for antibacterial activity of melanoidin.** We observed differences in the ability of melanoidins (22 conditions) to inhibit the growth of *L. monocytogenes* at 25°C for 24 h (Fig. 1). The highest antibacterial activity against *L. monocytogenes* was demonstrated by melanoidins derived from D-xylose-L-phenylalanine (Melanoidin-Phe) and D-xylose-L-proline (Melanoidin-Pro). The melanoidins derived from D-xylose-L-Leu, D-xylose-L-Phe, D-xylose-L-Pro, and D-glucose-L-Pro successfully inhibited the growth of *L. monocytogenes* for 24 h (absorbance change <0.1). The melanoidins derived from D-xylose-L-Ala, D-xylose-L-Arg, and D-xylose-L-Trp slightly inhibited (0.1 ≤ absorbance change <0.2) the growth of *L. monocytogenes*, and the remaining 15 melanoidins did not inhibit the growth of *L. monocytogenes* (absorbance change >0.2). The melanoidins derived from xylose at a concentration of 3.5 (absorbance at 420 nm) showed lower growth (smaller changes in absorbance at 595 nm) than those derived from glucose, regardless the kinds of amino acids used.

**Comparison of the effect of melanoidins on *L. monocytogenes* with the effect of nisin.** The melanoidins derived from D-xylose-L-Phe (Melanoidin-Phe) and D-xylose-L-Pro (Melanoidin-Pro), which showed the highest antibacterial activity against *L. monocytogenes* as shown in Fig. 1, were compared with nisin for their growth inhibition effects on *L. monocytogenes* (Fig. 2). Both melanoidins successfully inhibited *L. monocytogenes* growth over 48 h of incubation at 25°C. Both melanoidins showed a significant difference in *L. monocytogenes* growth inhibition compared to that of the negative control (i.e., without melanoidin or nisin). Both melanoidins exhibited no changes (no growth and no reduction) in viable cell numbers throughout the 48-h incubation period, regardless of the type of melanoidin tested. In contrast, the viable cell counts of *L. monocytogenes* decreased by approximately 2 log cycles from initial counts after 12 h of incubation in nisin solutions, regardless of their concentration. However, lower nisin concentration, such as 250 IU/mL, did not suppress the growth of *L. monocytogenes* for 48 h, and resulted in

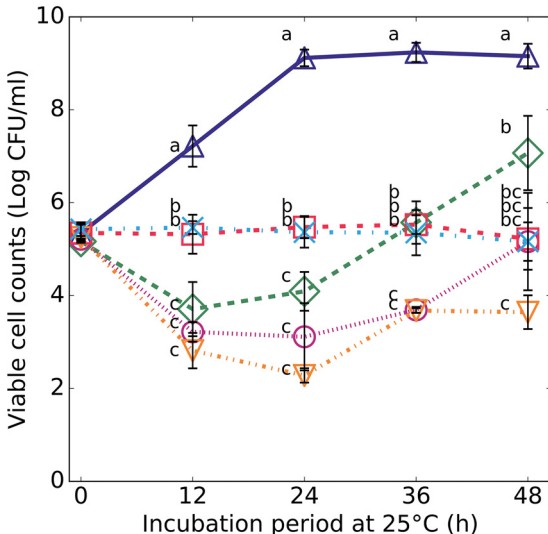

**FIG 2** Efficacy of melanoidin xylose-Phe (○), melanoidin xylose-Pro (×), nisin 250 IU/mL (◇), nisin 350 IU/mL (□), and nisin 400 IU/mL (▽) on the growth of *Listeria monocytogenes* ATCC 19111 in TSB medium. (△) indicates the negative control. Error bars represent the mean ± standard deviation of triplicate samples at each sampling point. Colony-counting data for each bacterium at each sampling interval were compared using the Tukey-Kramer multiple range test. *P* values of the results were used (*P* < 0.05); values with different letters represent significant differences.

an increase of approximately 2 log cycles after 48 h of incubation. The growth of *L. monocytogenes* in the 350 and 400 IU/mL nisin solutions was inhibited over the 48 h of incubation period compared with the initial viable cell numbers, although a trend of increase was apparently observed after the reduction during the first 24 h, regardless of nisin concentration.

**Antimicrobial spectra of the selected melanoidins.** Growth inhibition by melanoidin (melanoidin xylose-Phe and melanoidin xylose-Pro) against 11 species of bacteria was demonstrated at 25°C for 48 h (Table 1). The growth of *L. monocytogenes* and *Brevibacillus brevis* was successfully inhibited by both melanoidins for 48 h. Viable cell counts of *Brevibacillus brevis* decreased during the incubation period and then reached an undetectable level (<1 log CFU/mL). The growth of *B. cereus* was successfully inhibited by the xylose-Phe-based melanoidin, whereas it was not inhibited by the xylose-Pro-based melanoidin.

Growth of *E. coli*, *E. coli* O157:H7, and *S. enteritidis* was slightly inhibited by xylose-Pro-based melanoidin. Viable cell counts after 24 h of incubation at 25°C were significantly lower than those of the negative controls. The xylose-Pro-based melanoidin exhibited a strong antibacterial effect against *S.* Typhimurium. Viable cell counts of *S.* Typhimurium reached undetectable levels after 24 h of incubation at 25°C. However, the growth of other species of bacteria was not inhibited by either melanoidin.

**Antimicrobial activity of the melanoidins in solid conditions.** The antibacterial effects of melanoidin xylose-Phe and melanoidin xylose-Pro, which showed high effectiveness in liquid media, were tested on *L. monocytogenes* and *S.* Typhimurium in solid agar media at 25°C for 48 h (Table 2). While the growth of *L. monocytogenes* was slightly inhibited by xylose-Phe based melanoidin in solid medium, this growth was not significantly inhibited by xylose-Pro based melanoidin. The growth of *S.* Typhimurium was slightly inhibited by both melanoidins in solid media within 24 h compared with control sample. However, neither melanoidin inhibited the growth of *L. monocytogenes* and *S.* Typhimurium after 48 h of incubation in solid media at 25°C.

## DISCUSSION

Of all the amino acids used in the present study (Fig. 1), the melanoidins derived from D-xylose showed stronger antibacterial activity than those derived from glucose.

**TABLE 1** Effect of melanoidins on the growth of various bacteria at 25°C over 48 h in liquid culture media

| Bacterial strain | Incubation (h) | Viable cell counts (log CFU/mL)[a] | | |
| --- | --- | --- | --- | --- |
| | | | Melanoidin | |
| | | Negative control | Xyl-Phe[b] | Xyl-Pro[b] |
| **Gram-positive** | | | | |
| *B. brevis* NBRC 100599 | 0 | 5.96 ± 0.16 | 5.59 ± 0.31 | 5.68 ± 0.28 |
| | 24 | 7.76 ± 0.00 | ND[c] | 3.01 ± 0.15b |
| | 48 | 8.48 ± 0.05 | ND | ND |
| *B. cereus* ATCC 10987 | 0 | 4.93 ± 0.17 | 4.86 ± 0.14 | 4.92 ± 0.15 |
| | 24 | 7.96 ± 0.18a | 2.36 ± 1.13c | 5.72 ± 0.25b |
| | 48 | 8.09 ± 0.08a | 2.07 ± 0.25b | 7.97 ± 0.11a |
| *E. faecalis* ATCC 47077 | 0 | 5.29 ± 0.13 | 5.26 ± 0.04 | 5.25 ± 0.02 |
| | 24 | 9.46 ± 0.05a | 7.32 ± 0.54b | 7.75 ± 0.77b |
| | 48 | 8.82 ± 0.10a | 8.86 ± 0.09a | 8.85 ± 0.11a |
| *L. brevis* JCM 1059 | 0 | 5.33 ± 0.44 | 5.22 ± 0.17 | 5.23 ± 0.16 |
| | 24 | 7.44 ± 0.17a | 7.23 ± 0.09ab | 6.85 ± 0.15b |
| | 48 | 8.85 ± 0.03a | 8.49 ± 0.04b | 8.41 ± 0.12b |
| *L. fructivorans* NBRC 13954 | 0 | 5.03 ± 0.05 | 5.09 ± 0.05 | 5.14 ± 0.15 |
| | 24 | 6.29 ± 0.08a | 5.92 ± 0.12a | 6.02 ± 0.16a |
| | 48 | 7.54 ± 0.15a | 6.93 ± 0.04b | 7.29 ± 0.21ab |
| *L. monocytogenes* ATCC 19111 | 0 | 5.03 ± 0.26 | 5.09 ± 0.17 | 5.14 ± 0.15 |
| | 24 | 9.27 ± 0.16a | 5.43 ± 0.25b | 5.36 ± 0.33b |
| | 48 | 9.29 ± 0.27a | 5.33 ± 0.70b | 5.16 ± 1.05b |
| *S. epidermidis* ATCC 12228 | 0 | 4.57 ± 0.11 | 4.84 ± 0.05 | 4.78 ± 0.04 |
| | 24 | 8.98 ± 0.10a | 8.11 ± 0.20b | 7.57 ± 0.05c |
| | 48 | 9.05 ± 0.08a | 9.03 ± 0.04a | 9.00 ± 0.03a |
| **Gram-negative** | | | | |
| *E. coli* ATCC 25922 | 0 | 4.58 ± 0.21 | 4.63 ± 0.13 | 4.57 ± 0.08 |
| | 24 | 9.00 ± 0.05a | 8.96 ± 0.02a | 7.94 ± 0.09b |
| | 48 | 9.49 ± 0.16a | 9.44 ± 0.06a | 9.24 ± 0.02a |
| *E. coli* O157: H7 HIPH 11361 | 0 | 4.47 ± 0.03 | 4.47 ± 0.07 | 4.45 ± 0.12 |
| | 24 | 9.02 ± 0.03a | 8.95 ± 0.03a | 8.29 ± 0.14b |
| | 48 | 9.44 ± 0.01a | 9.13 ± 0.03b | 9.49 ± 0.14a |
| *S. enteritidis* RIMD 1933001 | 0 | 5.17 ± 0.30 | 5.11 ± 0.07 | 5.09 ± 0.05 |
| | 24 | 9.12 ± 0.11a | 8.93 ± 0.08a | 5.20 ± 0.11b |
| | 48 | 9.26 ± 0.17a | 9.22 ± 0.06a | 9.16 ± 0.09a |
| *S.* Typhimurium ATCC 29630 | 0 | 4.73 ± 0.03 | 4.64 ± 0.02 | 4.74 ± 0.04 |
| | 24 | 8.33 ± 0.08a | 4.70 ± 0.08b | ND |
| | 48 | 8.59 ± 0.15a | 8.14 ± 0.04b | ND |

[a]Each value is mean ± standard deviation ($n = 3$). Values with different letters in the same row (at the same sampling interval) represent significant differences ($P < 0.05$) using the Tukey-Kramer test.
[b]Xyl-Phe, xylose-phenylalanine; Xyl-Pro, xylose-proline.
[c]ND, not detected; detection limit is 1 log CFU/mL.

This result could be due to the reactivity of the reducing sugar. The Maillard reaction is affected by the type of reducing sugar used, based on its chemical structure. The higher the concentration of the open-chain form of the sugar, such as in pentose, the faster the browning proceeds (33). Thus, pentoses (such as xylose) promote a faster Maillard reaction when compared with hexoses with cyclic (closed-chain) form (such as glucose), with a 60-min reaction time; consequently, they might produce many melanoidins with higher antibacterial activity. Since it is possible that melanoidins with higher antibacterial activity could be produced from hexoses because of their longer reaction time, we should examine the effect of reaction time on the production of antibacterial melanoidins in the future.

Similarly, antimicrobial activity was dependent on the type of amino acid used. The melanoidins xylose-Phe and xylose-Pro had strong antibacterial activity against *L. monocytogenes*, while other combinations of melanoidin showed only slight or no activity over 24 h (Fig. 1). There are two possible reasons for the difference in antibacterial effects of the produced melanoidins. First, the structure of a melanoidin may be

Microbiology
Spectrum

**TABLE 2** Effect of melanoidins on the growth of *L. monocytogenes* and *S.* Typhimurium at 25°C over 48 h in solid culture media agar

| Bacterial strain | Incubation (h) | Viable cell counts (log CFU/mL)[a] | | |
| | | Negative control | Melanoidin | |
| | | | Xyl-Phe | Xyl-Pro |
| --- | --- | --- | --- | --- |
| *L. monocytogenes* ATCC19111 | 0 | 5.23 ± 0.43 | 5.61 ± 0.17 | 4.91 ± 0.22 |
| | 24 | 8.64 ± 0.08a | 6.99 ± 0.34c | 8.18 ± 0.08b |
| | 48 | 8.65 ± 0.20a | 8.33 ± 0.30a | 8.67 ± 0.04a |
| *S.* Typhimurium ATCC 29630 | 0 | 4.26 ± 0.27 | 4.25 ± 0.34 | 4.15 ± 0.27 |
| | 24 | 8.56 ± 0.38a | 7.67 ± 0.10b | 7.16 ± 0.05b |
| | 48 | 8.81 ± 0.16a | 8.84 ± 0.16a | 8.48 ± 0.20a |

[a]Each value represents mean ± standard deviation (*n* = 3). Values with different letters in the same row (at the same sampling interval) represent significant differences ($P <$ 0.05) as determined using the Tukey-Kramer test.

related to its antimicrobial activity. Melanoidin is a complex polymer and its exact structure is unknown, but the mechanism underlying its antimicrobial activity may be related to metal chelation, which is associated with damage to the bacterial membrane (26). We hypothesized that the chelating ability of melanoidin is dependent on the kind of the amino acid used to create it (Fig. 1). In chelation, polydentate and metal ions form coordinate bonds, and some of these grouped atoms (cyclic) sandwich metal atoms inside of them to become chelate complexes. A chelate complex is stable when it consists of a five- or six-membered ring structure, because the covalent bond angle is limited; L-Phe belongs to a six-membered ring, and L-Pro belongs to a five-membered ring. Although tryptophan has a ring structure, it is neither a five- nor six-membered ring, and its antibacterial activity in our study was not remarkable. Hence, we could consider that melanoidin with a stable chelating ability is produced using amino acids containing a five- or six-membered ring, and these melanoidins show high antibacterial activity.

Second, we hypothesized that antimicrobial peptides are incorporated into the melanoidin polymer. Peptides are polymers of approximately 2 to 50 amino acids, and the presence of peptide bonds within melanoidins has been suggested (34). Furthermore, some peptides have antibacterial properties (35, 36). Proline-rich peptides have shown antimicrobial effects against *Klebsiella pneumoniae* and *S. aureus*. They bind to the bacterial 70S ribosome and inhibit protein translation (37, 38). Furthermore, phenylalanine plays an essential role in some antimicrobial peptides (39) and may influence the antibacterial activity of cerein 8A, which is an antibacterial peptide (40). Based on these previous studies, we assume that melanoidin exhibits antibacterial activity either by chelation and/or due to the presence of antibacterial peptides.

Based on the results from our studies and previous reports, we focused on the melanoidins xylose-Phe and xylose-Pro and investigated their antibacterial spectra. As shown in Table 1, the antibacterial activity of these two melanoidins was stronger against Gram-positive bacteria than it was against Gram-negative bacteria. This trend agrees with an earlier study (41) in which Maillard reaction products more strongly suppressed the growth of Gram-positive bacteria, such as *Staphylococcus aureus* and *Bacillus subtilis*, than that of Gram-negative bacteria such as *E. coli* and *Salmonella*. These results may be due to the presence of an outer membrane, which is only found in Gram-negative bacteria. The outer membrane has a lipid bilayer and sugar chains which prevent the entry of hydrophilic and hydrophobic substances, respectively. These two mechanisms inhibit the antimicrobial effect of melanoidin against Gram-negative bacteria.

However, depending on the type of melanoidin, strong antibacterial activity was observed against both Gram-negative and Gram-positive bacteria. Melanoidin produced by the combination of xylose and proline (xylose-Pro) showed antibacterial activity against two *Salmonella* spp. In particular, viable cell counts of *S.* Typhimurium decreased below the detection limit (<1 log CFU/mL) after 24 h. Although other

researchers have reported the sensitivity of *S.* Typhimurium to Maillard reaction products (42, 43), the previous studies did not show an antibacterial effect of Maillard reaction products on *S.* Typhimurium. These differing results may be explained by the antimicrobial peptides present in melanoidin, as mentioned in the previous paragraph. Similar results have been reported for the proline-rich peptides that can inhibit *S.* Typhimurium (44).

In contrast, the two melanoidins in our study did not show antibacterial activity against *Lactobacillus brevis* and *Lactobacillus fructivorans*. Melanoidins are taken as daily dietary components in foods such as coffee and bread, and gut microbiota digest them. Because the gut microbiota is regularly exposed to some kind of melanoidin, the organisms in the gut microbiota might be resistant to melanoidins. Similar results were reported when testing digested glycoconjugates produced by the Maillard reaction: digested glycoconjugates stimulated the growth of *Lactobacillus brevis* and some other *Lactobacillus* spp. (45). Moreover, melanoidin is a type of prebiotic (16). The consideration that melanoidin is a carbon source for *Lactobacillus* spp. is, therefore, consistent with the findings in published reports.

Comparison of the results in Tables 1 and 2 shows that the sensitivity of *L. monocytogenes* and *S.* Typhimurium to melanoidin was much greater in liquid medium than in solid medium. This difference could be due to the accessibility of melanoidins to bacterial cells that cannot move freely. Similarly, the antibacterial effect of bacteriocins (such as nisin) and other inhibitory substances was weaker in solid media than in liquid media due to limited diffusion in solid media (5, 46, 47). Although bacterial cells could only contact a limited amount of melanoidins around themselves in solid media, they could continuously contact a greater amount of melanoidins in a volume of liquid media. Thus, for practical use in food products, it would be best to add melanoidin to liquid foods such as sauce, soup, and dressing.

As shown in Fig. 2, nisin reduced the amount of *L. monocytogenes* in a short time, within 2 days. However, although melanoidin inhibited the growth of *L. monocytogenes* for 2 days, it did not decrease its population. The results are almost consistent with previous reports on the effect of nisin on *L. monocytogenes* (4, 6, 7). The antibacterial activity of melanoidins after 48 h of incubation at 25°C was equivalent to the effect of nisin (350 IU/mL). Although the antimicrobial activity of melanoidin may be inferior to that of nisin based on the concentration required for inhibition, production cost and quality control for melanoidins may be more efficient compared with that of nisin; melanoidins can be made easily, by simply heating reducing sugar and amino acids in fluid. These results suggest the usefulness of the recently discovered melanoidins xylose-Phe and xylose-Pro as alternatives to conventional food preservatives.

The antibacterial activity of melanoidin does not exhibit universal effects as much as other antibacterial materials. Furthermore, melanoidins can influence the flavor of food depending on their concentration (48, 49). The higher the melanoidin concentration, the stronger the odor, and the color of melanoidins might be an adverse influence on food quality. Although the present study did not investigate the effects of flavor and color on food quality, these issues should be investigated in the future for practical application. Nonetheless, melanoidins exhibiting antibacterial activity have the potential for use as novel antibacterial agents by optimizing their usage concentration and/or their appropriate objectives.

## MATERIALS AND METHODS

**Bacterial strains.** *Bacillus cereus* ATCC 10987, *Brevibacillus brevis* NBRC 100599, *E. coli* ATCC 25922, *E. coli* O157:H7 HIPH 12361, *Enterococcus faecalis* ATCC 47077, *Lactobacillus brevis* JCM 1059, *L. fructivorans* NBRC 13954, *L. monocytogenes* ATCC 19111, *S. enteritidis* RIMD 1933001, *S.* Typhimurium ATCC 29630, and *S. epidermidis* ATCC 12221 were used (details on the strains used are listed in Table S1 in the supplemental material). The cultures were obtained from the American Type Culture Collection (Manassas, VA, USA), the National Institute of Technology and Evaluation (Tokyo, Japan), and the Research Institute for Microbial Diseases (Osaka, Japan).

All bacterial cultures were maintained at −80°C. Frozen *B. cereus*, *E. coli*, *E. coli* O157:H7, *L. monocytogenes*, *S. enteritidis*, and *S.* Typhimurium were streaked onto tryptic soy agar (TSA; Merck, Darmstadt,

**TABLE 3** Concentrations of reducing sugars and amino acids reacted at 121°C for 60 min to obtain melanoidins at concentrations of 3.5 ± 0.2 mM at an absorbance of 420 nm[a]

| | Examined concn. (mM) | |
| --- | --- | --- |
| Amino acid | D-Glucose | D-Xylose |
| L-Alanine | 65 | 40 |
| L-Arginine | 35 | 40 |
| L-Glutamine | 80 | 30 |
| L-Leucine | 60 | 35 |
| L-Methionine | 60 | 35 |
| L-Phenylalanine | 50 | 30 |
| L-Proline | 110 | 60 |
| L-Serine | 65 | 40 |
| L-Threonine | 85 | 40 |
| L-Tryptophan | 30 | 20 |
| L-Valine | 65 | 40 |

[a]Solutions were prepared in 1/15 M phosphate buffer (pH 7).

Germany) plates using a sterile platinum loop and incubated at 37°C for 24 h. Typical single colonies were selected, transferred to 5 mL of tryptic soy broth (TSB; Merck) in a sterile plastic tube, and then incubated at 37°C for 24 h. Frozen *L. brevis* and *L. fructivorans* were streaked onto MRS agar (Merck) plates and incubated at 30°C for 48 h. Typical single colonies were selected, transferred to 5 mL of MRS broth (Merck) in sterile plastic tubes, and then incubated at 37°C for 24 h.

**Preparation of melanoidin.** Twenty-two water-soluble melanoidins were produced by the Maillard reaction system based on a previous study (50, 51). L-Alanine, L-arginine, L-glutamine, L-leucine, L-methionine, L-phenylalanine, L-proline, L-threonine, L-valine, L-serine, and L-tryptophan (FUJIFILM Wako Pure Chemical Corporation, Osaka, Japan) were used as amino acids for the reaction substrate. D-Glucose (Kanto Chemical Corporation, Tokyo, Japan) and D-xylose (FUJIFILM Wako Pure Chemical Corporation) were used as reducing sugars for the reaction substrate. The molar ratio of reaction substrates (reducing sugar: amino acid) was fixed at 1:0 in phosphate buffer (1/15 M, [pH 7.0]) regardless of the types of amino acid and reducing sugar used. All the mixtures were heated at 121°C for 60 min in glass tubes by autoclaving and, immediately after heating, the reactants were immersed in an iced water bath to prevent excessive progress of the Maillard reaction. The reactants obtained by the heating process described above were regarded as melanoidins in this study.

Absorbance at 420 nm was used to determine the concentration of melanoidin, because the degree of browning correlates with melanoidin concentration (50, 52, 53). The absorbance of the reactants at 420 nm was measured using a spectrophotometer (UV-1700, Shimadzu, Kyoto, Japan). To compare the differences in antibacterial activity resulting from the various reaction substrates, the concentration of melanoidin (absorbance at 420 nm) after production was adjusted to 3.5 ± 0.2, by changing the molar concentration of the amino acid and the reducing sugar and maintaining a molar ratio of 1.0. The molar concentrations of all combinations of reaction substrates are listed in Table 3.

**Effect of various melanoidins on the growth of *L. monocytogenes*.** The growth inhibition activities of the 22 melanoidins produced by the combination of two reducing sugars and 11 amino acids was assessed by determining the growth of *L. monocytogenes* at 25°C for 24 h, defined as changes in absorbance at 595 nm (iMark microplate reader, Bio-Rad Laboratories, Hercules, CA, USA) during an incubation period at 25°C (54, 55). The cell suspension of *L. monocytogenes*, prepared as described above, was used as a model bacterium. Melanoidins (100 μL) and TSB (100 μL) were dispensed into a 96-well microplate and inoculated with *L. monocytogenes* (20 μL) to obtain ~105 CFU/well. For negative controls, the growth of *L. monocytogenes* in the solution without melanoidins was also determined by adding 0.1% peptone water (100 μL) instead of the melanoidin solution. Changes in absorbance-at-595-nm values for all tested solutions during incubation at 25°C for 24 h were determined independently, in triplicate, and the values were averaged to represent the absorbance for each condition. This procedure was intended to screen and identify apparent growth inhibition conditions for further quantitative growth evaluation as described below.

**Comparison of melanoidin antibacterial activity with that of nisin.** To evaluate their relative antibacterial activity, we compared the antibacterial activity of melanoidins with that of nisin against *L. monocytogenes*. The melanoidins xylose-Phe and xylose-Pro were prepared as described in Results. Nisin (San-Ei Gen F.F.I. Inc., Osaka, Japan) was dissolved in 0.1% peptone water at concentrations of 250, 350, and 400 IU/mL in TSB. Stationary-phase *L. monocytogenes* (200 μL, 10⁵ to 10⁶ CFU/mL) was added to the test solutions with or without nisin or melanoidins (10 mL). All samples were incubated at 25°C for 48 h and sampled at 12-h intervals.

**Antibacterial effects of specific melanoidins on various bacterial species in liquid media.** We evaluated the antibacterial spectra of two types of melanoidins (derived from D-xylose with L-Phe and D-xylose with L-Pro, as described previously) against various bacterial species, including Gram-positives of *Bacillus cereus* (ATCC 10987), *Brevibacillus brevis* (NBRC 100599), *Enterococcus faecalis* (ATCC 47077), *Lactobacillus brevis* (JCM 1059), *Lactobacillus fructivorans* (NBRC 13954), *Listeria monocytogenes* (ATCC 19111), *Staphylococcus epidermidis* (ATCC 12228), and Gram-negatives of *Escherichia coli* (ATCC 25922), *E.*

*coli* O157:H7 (HIPH 11361), *Salmonella enteritidis* (RIMD 1933001), *Salmonella* Typhimurium (ATCC 29630).

Each melanoidin for which absorbance at 420 nm was 3.5 ± 0.2 (500 $\mu$L), together with TSB or MRS broth (500 $\mu$L), was mixed in sterile 1.5-mL sampling tubes and inoculated with each bacterial suspension (20 $\mu$L) to obtain $10^5$ to $10^6$ CFU/mL. Viable bacterial cell counts were determined by direct plate counting at 0, 24, and 48 h post-incubation at 25°C. An aliquot of 100 $\mu$L bacterial suspension was taken at each sampling interval and serially diluted with 0.1% peptone water, and 100 $\mu$L of diluted samples were plated onto TSA; except for lactic acid bacteria (*E. faecalis*, *L. brevis*, and *L. fructivorans*), which were plated onto MRS agar. The inoculated TSA and MRS agar plates were incubated at 37°C for 24 h and 30°C for 48 h, respectively.

**Antimicrobial assay in solid media.** We evaluated the antibacterial activities of two types of melanoidins (derived from D-xylose with L-Phe and D-xylose with L-Pro) in solid medium to investigate whether melanoidin shows antibacterial activity under both solid and liquid conditions. Melanoidins in solid conditions were prepared based on the procedure described previously. To prepare the solid medium, 1.5% agar was added to the media, with or without melanoidin.

*L. monocytogenes* and *S.* Typhimurium were used as representative bacterial strains based on the results of the effect on liquid melanoidins. The pregrown bacterial suspension was mixed with culture media using the pour-plate method. Plate samples were incubated at 25°C for 48 h, and petri dishes were removed from the incubator at 0, 24, and 48 h afterwards. The agar culture was removed from the petri dish and mashed with 50 mL of 0.1% peptone water for 60 s, using a homogenizer (ELMEX, Tokyo, Japan). The mashed samples were serially diluted, and 100 $\mu$L of the sample was applied to TSA and incubated at 37°C for 24 h.

**Statistical analysis.** Changes in viable cell counts for all the tested bacteria, with or without melanoidin, were determined three times in independent experiments (see previous subsections). The colony-counting data for the triplicate samples of each bacterium at each sampling interval were transformed to log CFU/mL. Statistically significant differences ($P < 0.05$) in absorbance changes for 24 h among the 22 types of melanoidins and the negative control, and statistically significant differences ($P < 0.05$) in viable cell counts at each sampling time, were determined using the Tukey-Kramer multiple range test. Statistical analysis was conducted using the Pandas package (v. 1.2.1) of Python (v. 3.8.1) managed by the Python Software Foundation

**Data availability.** The source data for the results, as shown in Figure 1 and 2, are summarized in the supplemental file. In addition, data for changes in bacterial numbers are illustrated in Tables 1 and 2.

## SUPPLEMENTAL MATERIAL

Supplemental material is available online only.

**SUPPLEMENTAL FILE 1**, PDF file, 0.1 MB.

## ACKNOWLEDGMENTS

This study was supported by a research grant from the Tojuro Iijima Foundation for Food Science and Technology.

We thank Editage (www.editage.com) for English language editing.

S. Kukuminato contributed Investigation, Visualization, Writing – Original Draft Preparation; K. Koyama contributed Methodology, Software, Writing – Reviewing and Editing; S. Koseki contributed Conceptualization, Supervision, Writing– Reviewing and Editing.

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
