## [Reviewer comments · Microbiology Spectrum]

Microbiology Spectrum

Antibacterial properties of melanoidins produced from various combinations of Maillard reaction against pathogenic bacteria

Satoshi Kukuminato, Kento Koyama, and Shige Koseki

Corresponding Author(s): Shige Koseki, Hokkaido University

Review Timeline:

Submission Date:	August 3, 2021
Editorial Decision:	October 26, 2021
Revision Received:	November 8, 2021
Accepted:	November 14, 2021

Editor: Luca Cocolin

Reviewer(s): Disclosure of reviewer identity is with reference to reviewer comments included in decision letter(s). The following individuals involved in review of your submission have agreed to reveal their identity: Donald W. Schaffner (Reviewer #1)

Transaction Report:

DOI: <https://doi.org/10.1128/spectrum.01142-21>

October 26, 2021

Prof. Shige Koseki
Hokkaido University
Graduate School of Agricultural Science
Kita 9 Nishi 9 Kita-ku
Sapporo 060-8589
Japan

Re: Spectrum01142-21 (Antibacterial properties of melanoidins produced from various combinations of Maillard reaction against pathogenic bacteria)

Dear Prof. Shige Koseki:

Thank you for submitting your manuscript to Microbiology Spectrum. When submitting the revised version of your paper, please provide (1) point-by-point responses to the issues raised by the reviewers as file type "Response to Reviewers," not in your cover letter, and (2) a PDF file that indicates the changes from the original submission (by highlighting or underlining the changes) as file type "Marked Up Manuscript - For Review Only". Please use this link to submit your revised manuscript - we strongly recommend that you submit your paper within the next 60 days or reach out to me. Detailed information on submitting your revised paper are below.

Link Not Available

Sincerely,

Luca Cocolin

Journals Department
Reviewer comments:

Reviewer #1 (Comments for the Author):

The authors present an interesting analysis. Although they have used an editing service the manuscript will still need careful review for correct English usage. The work is generally sound although there are a number of issues that will require attention prior to publication including a number of paragraphs in the discussion that do not discuss the work in the context of appropriate published literature.

Page 7:

Content: " showed significantly difference"

Comment: Minor grammatical error, should be "a significant difference".

Page 9:

Content: " In contrast, the growth of *S. Typhimurium* was inhibited by both the melanoidins in solid media within 24 h. "

Comment: While it is true that the growth in the test sample is less than in the control sample, even in the test sample there is a three log increase over the starting concentration, so I'm not sure that the word "inhibited" is correct here.

Page 9:

Content: "result would be due to the reactivity of the reducing sugar."

Comment: Would be, or could be?

Page 9:

Content: "might produce many melanoidins that have higher antibacterial activity."

Comment: I see the authors point, however if a reaction with a hexose was allowed to proceed for a longer time, would this not at some point have the same antibacterial activity?

Page 9:

Content: "structure of the amino acid (Fig. 1)."

Comment: I am not sure why the authors reference figure one here, as it has nothing to do with the structure of the amino acid.

Page 10:

Content: " This agrees with an earlier study (Einarsson et al., 1983). "

Comment: Please explain exactly what this earlier study showed.

Page 11:

Content: "These results imply that some Lactobacillus spp. may use melanoidin as a carbon source."

Comment: I'm not sure that it implies this. Perhaps the authors could elaborate or explain further.

Page 11:

Content: "these bacteria may be resistant to melanoidin."

Comment: Or conversely, these organisms exist in the gut microbiota because they are resistant to the compound.

Page 11:

Content: "melanoidin may become a prebiotic"

Comment: Why "may become", why not "is"?

Page 11:

Content: "Comparison of the results in Table 1 and Table 2 shows that the sensitivity of L."

Comment: this paragraph does not discuss the results in the context of other literature, therefore it is not appropriate for the discussion. It would be very interesting for the authors to discuss whether this difference between solid and liquid systems is seen with other preservatives like Nicin. I'm sure this must have been studied.

Page 12:

Content: "This hypothesis might support the results that the bacterial growth was slightly inhibited within a short period such as 24 h (Table 2), which meant that the limited amount of melanoidins around the bacterial cells would work."

Comment: These are interesting observations but again this paragraph really needs to be written so that it belongs in the discussion.

Page 12:

Content: "As shown in Fig. 2, nisin reduced the number of L. monocytogenes in a short time, within 2 days."

Comment: This is another paragraph which does not really belong in the discussion since it does not discuss the research in the context of other research. Do the authors findings with nisin match what others have reported? This would be a place to discuss this.

Page 12:

Content: "Furthermore, melanoidins would influence the color and flavor of food depending on the concentration."

Comment: Again this is another paragraph which does not discuss other published literature. How do the concentrations studied by the authors relate to other research? Have there been any studies that look at consumer acceptability with respect to flavor and color?

Page 30:

Content: "negative control"

Comment: The negative control on the plot has dots but the legend in the figure shows no dots. This should be made consistent.

Staff Comments:

Preparing Revision Guidelines

Please return the manuscript within 60 days; if you cannot complete the modification within this time period, please contact me. If you do not wish to modify the manuscript and prefer to submit it to another journal, please notify me of your decision immediately so that the manuscript may be formally withdrawn from consideration by Microbiology Spectrum.

November 7, 2021

Spectrum01142-21

Title: Antibacterial properties of melanoidins produced from various combinations of Maillard reaction against pathogenic bacteria

Dear Reviewer#1:

Thank you for your critical reviewing. We appreciate reviewer's suggestions and comments. We have responded your comments and revised the manuscript accordingly. The responses for the comments are written in **blue text** and the revised parts are highlighted **in yellow**.

Reviewer #1 (Comments for the Author):

The authors present an interesting analysis. Although they have used an editing service the manuscript will still need careful review for correct English usage. The work is generally sound although there are a number of issues that will require attention prior to publication including a number of paragraphs in the discussion that do not discuss the work in the context of appropriate published literature.

Page 7:

Content: " showed significantly difference"

Comment: Minor grammatical error, should be "a significant difference".

⇒ Thank you for the comment. We have revised accordingly as follows:

"**showed a significant difference**" Please see Line 151 in the revised manuscript.

Page 9:

Content: " In contrast, the growth of S. Typhimurium was inhibited by both the melanoidins in solid media within 24 h. "

Comment: While it is true that the growth in the test sample is less than in the control sample, even in the test sample there is a three log increase over the starting concentration, so I'm not sure that the word "inhibited" is correct here.

⇒ Thank you for the comment. As the reviewer#1 indicated, this expression was inappropriate. We have revised the description as follows:

"**The growth of S. Typhimurium was slightly inhibited by both the melanoidins in solid media within 24 h compared with control sample.**"

Please see Lines 185 -186 in the revised manuscript.

Page 9:

Content: "result would be due to the reactivity of the reducing sugar."

Comment: Would be, or could be?

⇒ Thank you for the comment. According to the Reviewer#1's suggestion, we have revised accordingly as follows:

"**This result could be due to the reactivity of the reducing sugar.**"

Please see Lines 194 in the revised manuscript.

Page 9:

Content: "might produce many melanoidins that have higher antibacterial activity."

Comment: I see the authors point, however if a reaction with a hexose was allowed to proceed for a longer time, would this not at some point have the same antibacterial activity?

⇒ Thank you for the comment. As the reviewer#1 mentioned, there may be a possibility. However, since we did not examine the reaction longer than 60 min, we could not say more than possibility. But, we have revised as follows:

"Thus, pentoses such as xylose promote Maillard reaction compared with hexoses such as glucose within 60 min reaction time and consequently might produce many melanoidins that have higher antibacterial activity. Since there might be a possibility that the melanoidins with higher antibacterial activity could be produced from hexoses by much longer reaction time, we should examine the effect of reaction time on the production of antibacterial melanoidins in the future."

Please see Lines 198-202 in the revised manuscript.

Page 9:

Content: "structure of the amino acid (Fig. 1)."

Comment: I am not sure why the authors reference figure one here, as it has nothing to do with the structure of the amino acid.

⇒ This was our mistake. We have revised as follows:

"dependent on the kinds of the amino acid.

Please see Line 212 in the revised manuscript.

Page 10:

Content: " This agrees with an earlier study (Einarsson et al., 1983). "

Comment: Please explain exactly what this earlier study showed.

⇒ Thank you for the comment. The previous study showed higher antibacterial effect on Gram-positives than Gram-negatives. We have revised as follows:

This trend agrees with an earlier study (Einarsson et al., 1983) in which Maillard reaction products suppressed growth of Gram-positive bacteria such as *Staphylococcus aureus* and *Bacillus subtilis* than those of Gram-negative bacteria such as *E. coli* and *Salmonella*.

Please see Lines 235 -238 in the revised manuscript.

Page 11:

Content: "These results imply that some *Lactobacillus* spp. may use melanoidin as a carbon source."

Comment: I'm not sure that it implies this. Perhaps the authors could elaborate or explain further.

⇒ Thank you for the comment. This sentence is inappropriate for the context, we have deleted the sentence.

Page 11:

Content: "these bacteria may be resistant to melanoidin."

Comment: Or conversely, these organisms exist in the gut microbiota because they are resistant to the compound.

⇒ Thank you for the comment. According to the Reviewer#1's comment, the sentence has been revised as follows:

"Because the gut microbiota is regularly exposed to a kind of melanoidin, these organisms exist in the gut microbiota because they are resistant to the Melanoidins."

Please see Lines 256-258 in the revised manuscript.

Page 11:

Content: "melanoidin may become a prebiotic"

Comment: Why "may become", why not "is"?

⇒ Thank you for the comment. We have corrected accordingly.

"melanoidin is a kind of prebiotic"

Please see Line 261 in the revised manuscript.

Page 11:

Content: "Comparison of the results in Table 1 and Table 2 shows that the sensitivity of L."

Comment: this paragraph does not discuss the results in the context of other literature, therefore it is not appropriate for the discussion. It would be very interesting for the authors to discuss whether this difference between solid and liquid systems is seen with other preservatives like Nicin. I'm sure this must have been studied.

⇒ Thank you for the comment. We have revised the paragraph citing the literature as follows:

"Similarly, the antibacterial effect of bacteriocin including nisin or other inhibitory substance in solid media was weaker than that in liquid media due to the limited diffusion in solid media (Delves-Broughton et al., 1996; Blom et al., 1997; Coma et al., 2001)."

Please see Lines 267-270 in the revised manuscript.

Page 12:

Content: "This hypothesis might support the results that the bacterial growth was slightly inhibited within a short period such as 24 h (Table 2), which meant that the limited amount of melanoidins around the bacterial cells would work."

Comment: These are interesting observations but again this paragraph really needs to be written so that it belongs in the discussion.

⇒ Thank you for the comment. We have deleted the sentence to avoid over-speculation.

Page 12:

Content: "As shown in Fig. 2, nisin reduced the number of L. monocytogenes in a short time, within 2 days."

Comment: This is another paragraph which does not really belong in the discussion since it does not discuss the research in the context of other research. Do the authors

findings with nicin match what others have reported? This would be a place to discuss this.

⇒ Thank you for the comment. We have cited related literature in the text.

“The results are consistent with the previous reports on the effect of nisin on *L. monocytogenes* (Kierończyk et al., 2020; Punyauppa-path et al., 2015; Ukuku et al., 1997).”

Please see Lines 277-279 in the revised manuscript.

Page 12:

Content: "Furthermore, melanoidins would influence the color and flavor of food depending on the concentration."

Comment: Again this is another paragraph which does not discuss other published literature. How do the concentrations studied by the authors relate to other research? Have there been any studies that look at consumer acceptability with respect to flavor and color?

⇒ Thank you for the comment. We have cited relevant literature on flavor and describe the limitation of the present study as follows:

“Furthermore, melanoidins would influence the flavor of food depending on the concentration (Adams et al., 2005; Hofmann et al., 2001). The higher the concentration of melanoidins, the higher the intensity of odor and color of melanoidins would be adversely influenced on food quality. Although the present study did not investigate the effect of flavor and color on food quality, these issues should be investigated in the future for practical application.”

Please see Lines 288-293 in the revised manuscript.

Page 30:

Content: "negative control"

Comment: The negative control on the plot has dots but the legend in the figure shows no dots. This should be made consistent.

⇒ Thank you for the comment. We have revised the Figure 1.

Thank you again for the Reviewer#1's reviewing and comments.

November 14, 2021

Prof. Shige Koseki
Hokkaido University
Graduate School of Agricultural Science
Kita 9 Nishi 9 Kita-ku
Sapporo 060-8589
Japan

Re: Spectrum01142-21R1 (Antibacterial properties of melanoidins produced from various combinations of Maillard reaction against pathogenic bacteria)

Dear Prof. Shige Koseki:

Your manuscript has been accepted, and I am forwarding it to the ASM Journals Department for publication. You will be notified when your proofs are ready to be viewed.

Sincerely,

Luca Cocolin
Editor, Microbiology Spectrum

Journals Department
Supplemental Table 1: Accept